# An Efficient Metal-Free Oxidative Esterification and Amination of Benzyl C–H Bond

**DOI:** 10.3390/molecules25071527

**Published:** 2020-03-27

**Authors:** Saiwen Liu, Ru Chen, Guowen He, Jin Zhang

**Affiliations:** 1College of Materials and Chemical Engineering, Hunan City University, Yiyang 413000, China; zhongyihgw@163.com; 2Yiyang Agriculture Products Quality Detect Center, Yiyang 413000, China; chenrudaxia@163.com

**Keywords:** esterification, amination, DDQ, diarylmethanes, benzyl ester

## Abstract

An esterification and amination of benzylic C–H bonds was developed by using 2,3-dichloro-5,6-dicyano-1,4-benzoquinone (DDQ) under metal- and iodide-free conditions. Both carboxylic acids and amines could be used as ideal coupling partners for the oxidative coupling reactions with various diarylmethanes. A close to equal amount of coupling reagents was enough to afford the product in good to high yields.

## 1. Introduction

Benzyl esters, which play a key role in many biologically active compounds, can be traditionally prepared by condensation of the corresponding benzyl alcohols with carboxylic acids in the presence of condensation reagents or their activated derivatives such as acyl chlorides in the presence of bases [1,2]. They also serve as the essential protecting groups in amino acids and their derivatives [3]. Therefore, development of efficient methods for rapid construction of benzyl esters has gained much attention and great progress has been made during the past several decades. Recently, the cleavage and functionalization of C–H bonds is of increasing interest for both academia and industry. Generally, the transformation mainly relies on transition metals [4,5,6]. This strategy has also been successfully employed for the direct esterification and amination of C–H bonds including benzylic C–H bonds under oxidative reaction conditions. Various transition-metals efficiently catalyzed the direct esterification [7,8] and amination [9,10,11] of C–H bonds. Due to the toxicity and cost of transition-metal catalysts, increasing interest has been paid on metal-free esterifications of benzylic C–H bonds.

Since Sinha and co-workers reported a 2,3-dichloro-5,6-dicyano-1,4-benzoquinone (DDQ) promoted acetoxylation of arylalkanes under sonochemical activation conditions in the absence of metal catalyst [12], various efficient esterification methods have been developed under metal-free conditions. The combination of quaternary ammonium iodides with strong oxidants was found to be an effective strategy for direct esterification and amination of benzylic C–H bonds [13,14,15,16]. Recently, Shen and co-workers reported the C–O coupling of C–H with carboxylic acids catalyzed by DDQ and *tert*-butyl nitrite using oxygen as the terminal oxidant [17]. However, large excess of carboxylic acids or amines are necessary to get satisfied reaction yields [18]. The development of esterification reactions using a near equal amount of reactants under metal-free conditions would be highly desirable. Herein, we report a DDQ promoted esterification and amination of diarylmethanes using only 1.5 equiv. of carboxylic acids and amines without the use of quaternary ammonium iodide or metal catalyst. The chloro-containing solvent plays an important role for this efficient transformation.

## 2. Results

### 2.1. Optimization of Reaction Conditions for Synthesis of Benzhydryl Benzoate **3a**

We began our study by examining the reaction of diphenylmethane (**1a**) and with 1.5 equiv. of benzoic acid (**2a**) using DDQ as oxidant in 1,2-dichloroethane (DCE) at 100 °C under an atmosphere of argon. DDQ showed very good activity and the desired product benzhydryl benzoate (**3a**) was obtained in 95% yield as determined by GC (Table 1, entry 1). The oxidant played an important role for this kind of reaction. Other common oxidants, such as benzoquinone, peroxides and hydrogen peroxide, were ineffective and no desired product was observed (entries 2–6). Solvents also played an important role for the direct esterification. The reactions in chloro-containing solvents, such as chloroform, dichloromethane, and tetrachloroethane, all showed very good reactivity (entries 7–9). Other solvents, such as ethyl acetate, acetonitrile, and water, all decreased the reaction yields significantly (entries 10–13). Decreasing the reaction temperature decreased the reaction yield (entry 14). The reaction yields decreased when the reactions were carried out under oxygen and air (entries 15 and 16). Good yield still could be achieved when an equal amount of **2a** was used (entry 17).

### 2.2. Substrate Scope for the Carboxylic Acids and Amines

With the optimized reaction conditions established, the scope of the reaction with respect to diphenylmethane (**1a**) and various carboxylic acids (**2**) was investigated (Table 2). The reactions with aromatic carboxylic acids bearing electron-donating groups and electron-withdrawing substituents at the *para* position proceeded smoothly to give the desired products in good yields (**3b**–**3e**). The position of the substituents on the phenyl ring of benzoic acids affected the reaction yield slightly, and the use of *ortho*-substituted benzoic acids also afforded the desired products in good yields. Under the optimized conditions, halogen substituents were all well tolerated (**3e**–**3h**). Good yield was obtained when 2-bromobenzoic acid was used, and the desired product **3h** was achieved in 83% yield. More bulky substrates, such as 2,4,6-trimethylbenzoic acid, also efficiently reacted with **1a** and gave the product in 85% yield (**3j**). Notably, high yield was achieved when hetero aromatic carboxylic acids, such as 3-methylthiophene-2-carboxylic acid, reacted with **1a**, and the desired product **3k** was obtained in 88% yield. To be our delight, aliphatic carboxylic acids, including formic acid, are also good coupling reagents for this kind of transformation, and the desired esters were formed in high yields (**3m**–**3p**). In addition to carboxylic acids, amines could also be used as ideal coupling partners for the oxidative coupling reaction. Diphenylmethane smoothly coupled with benzamide and gave the aminated product **4a** in 62% yield. Moderate to good yields were obtained when sulfonamides were employed for the coupling reaction (**4b**–**4d**).

### 2.3. Substrate Scope for the Diarylmethanes

To further explore the scope of the reaction, various diarylmethanes (**1**) were employed to react with **2a** under the optimized conditions (Table 3). A series of functional groups, including methyl, *tert*-butyl, fluoro, and chloro, were well tolerated under the optimized conditions (**3t**–**3x**). The position of the substituents on the phenyl ring of diphenylmethane only slightly affected the reaction yield (**3q**–**3s**). The desired product **3y** was obtained in 40% yield when 1-benzyl-4-nitrobenzene was used. Unfortunately, this reaction condition was only suitable for diarylmethane substrates. Other benzylic substrates, such as toluene and ethylbenzene, were not reactive under the optimized reaction conditions.

### 2.4. Mechanism

To gather more information about the reaction mechanism, the radical scavenger 2,2,6,6-tetramethyl-piperidine-1-oxyl (TEMPO) was added. No product was obtained from the reaction between diphenylmethane and benzoic acid with the influence of two equivalents of TEMPO (Scheme 1). This indicates that the present reaction process may involve radical intermediates.

A plausible mechanism to rationalize this transformation is illustrated in Scheme 2. The reaction of DDQ with diphenylmethane **1a** generates a benzyl radical **I** and DDQH**^·^** via hydrogen atom transfer (HAT). The radicals continue to be transformed into the corresponding benzyl cation and DDQH anion via single electron transfer (SET) [17,18]. Finally, the benzoic acid **2a** of the proton is abstracted by DDQH^–^ and reacts with the cation **II** to obtain the desired product **3a** and reduced hydroquinone DDQH_2_. However, the reason why the reaction only works well in chloro-containing solvents is not clear at this stage.

## 3. Materials and Methods

### 3.1. General Information

All experiments were carried out under an atmosphere of argon. Flash column chromatography was performed over silica gel 48–75 μm. ^1^H-NMR and ^13^C-NMR spectra were recorded on a Bruker-AV (400 and 100 MHz, respectively) instrument (Billerica, MA, USA) internally referenced to SiMe_4_ or chloroform signals. MS analyses were performed on an Agilent 5975 GC-MS instrument (EI) (Santa Clara, CA, USA). The new compounds were characterized by ^1^H NMR, ^13^C NMR, MS, and HRMS. The structure of known compounds were further corroborated by comparing their ^1^H-NMR, ^13^C-NMR, and MS data with those in the literature. All reagents were used as received from commercial sources without further purification. Diarylmethane derivatives **1** were synthesized based on relevant literatures [19].

### 3.2. General Procedure for the Synthesis of Benzhydryl Benzoate

An oven-dried pressure tube (10 mL) was charged with diphenylmethane (**1a**, 33.4 μL, 0.2 mmol), benzoic acid (**2a**, 36.6 mg, 0.3 mmol), DDQ (54.5 mg, 0.24 mmol), and DCE (0.6 mL). The reaction vessel was flushed with argon and sealed. The resulting solution was heated to 100 °C for 8 h. After cooling to room temperature, the volatiles were removed under vacuum and the residue was purified by column chromatography (silica gel, petroleum ether/ethyl acetate = 9:1) to give **3a** as white solid; yield: 46.1 mg (80%). (NMR spectra for all compounds shown in the Appendix A).

### 3.3. Product Characterization

*Benzhydryl benzoate* (**3a**) [17]: white solid, 80% yield (46.1 mg), ^1^H NMR (400 MHz, CDCl_3_, ppm) *δ* 8.15 (d, *J* = 7.6 Hz, 2H), 7.58 (d, *J* = 7.0 Hz, 1H), 7.43–7.46 (m, 6H), 7.36 (t, *J* = 7.4 Hz, 4H), 7.30 (d, *J* = 7.2 Hz, 2H), 7.12 (s, 1H); ^13^C NMR (100 MHz, CDCl_3_, ppm) *δ* 165.6, 140.4, 133.1, 130.1, 129.8, 128.6, 128.5, 128.0, 127.2, 77.5. MS (EI) *m/z* (%) 288, 183, 166 (100), 152, 105, 77.

*Benzhydryl 4-methylbenzoate* (**3b**) [20]: white solid, 80% yield (48.3 mg), ^1^H NMR (400 MHz, CDCl_3_, ppm) *δ* 8.03 (d, *J* = 8.0 Hz, 2H), 7.27–7.44 (m, 12H), 7.10 (s, 1H), 2.42 (s, 3H); ^13^C NMR (100 MHz, CDCl_3_, ppm) *δ* 165.7, 143.9, 140.5, 129.9, 129.2, 128.6, 128.0, 127.6, 127.2, 77.3, 21.7. MS (EI) *m/z* (%) 302, 166 (100), 152, 119, 91.

*Benzhydryl 3-methylbenzoate* (**3c**): White solid, 84% yield (50.7 mg), ^1^H NMR (400 MHz, CDCl_3_, ppm) *δ* 7.94 (m, 2H), 7.30–7.45 (m, 12H), 7.12 (s, 1H), 2.41 (s, 3H); ^13^C NMR (100 MHz, CDCl_3_, ppm) *δ* 165.8, 140.4, 138.3, 134.0, 132.5, 130.3, 128.6, 128.4, 128.0, 127.2, 127.0, 77.4, 21.4; MS (EI) *m/z* (%) 302, 166 (100), 152, 119, 91; HRMS calcd. for: C_21_H_18_NaO_2_ [M + Na]^+^: 325.1201, found *m/z* 325.1205.

*Benzhydryl 2-methylbenzoate* (**3d**) [21]: colorless liquid, 82% yield (49.5 mg), ^1^H NMR (400 MHz, CDCl_3_, ppm) *δ* 8.08 (d, *J* = 7.6 Hz, 1H), 7.31–7.46 (m, 13H), 7.01 (s, 1H), 2.61 (s, 3H); ^13^C NMR (100 MHz, CDCl_3_, ppm) *δ* 166.4, 140.7, 140.5, 132.2, 131.9, 130.8, 129.5, 128.6, 128.0, 127.2, 125.9, 77.4, 22.0; MS (EI) *m/z* (%) 302, 166 (100), 152, 119, 91.

*Benzhydryl 4-fluorobenzoate* (**3e**) [22]: colorless liquid, 70% yield (42.9 mg), ^1^H NMR (400 MHz, CDCl_3_, ppm) *δ* 8.10 (t, *J* = 6.8 Hz, 2H), 7.20–7.37 (m, 10H), 7.04–7.09 (m, 3H); ^13^C NMR (100 MHz, CDCl_3_, ppm) *δ* 165.9 (d, *J* = 259.2 Hz), 164.7, 140.2, 132.4 (d, *J* = 9.3 Hz), 130.0, 128.6, 128.0, 127.2, 115.6 (d, *J* = 21.9 Hz), 77.7; MS (EI) *m/z* (%) 306, 183, 166 (100), 152, 123, 95.

*Benzhydryl 2-fluorobenzoate* (**3f**) [23]: colorless liquid, 82% yield (50.2 mg), ^1^H NMR (400 MHz, CDCl_3_, ppm) *δ* 7.91 (d, *J* = 7.2 Hz, 1H), 7.69 (d, *J* = 7.6 Hz, 1H), 7.47 (d, *J* = 7.2 Hz, 4H), 7.32–7.40 (m, 8H), 7.14 (s, 1H); ^13^C NMR (100 MHz, CDCl_3_, ppm) *δ* 163.5, 162.2 (d, *J* = 258.9 Hz), 140.2, 134.7 (d, *J* = 9.0 Hz), 132.4, 128.6, 128.0, 127.2, 124.1 (d, *J* = 4.0 Hz), 118.8, 117.1 (d, *J* = 22.4 Hz), 78.0; MS (EI) *m/z* (%) 306, 183, 166 (100), 152, 123, 95.

*Benzhydryl 2-chlorobenzoate* (**3g**) [23]: colorless liquid, 81% yield (52.3 mg), ^1^H NMR (400 MHz, CDCl_3_, ppm) *δ* 7.93 (d, *J* = 7.6 Hz, 1H), 7.30–7.46 (m, 13H), 7.13 (s, 1H); ^13^C NMR (100 MHz, CDCl_3_, ppm) *δ* 164.6, 140.0, 134.1, 132.7, 131.7, 131.2, 130.1, 128.6, 128.0, 127.3, 126.6, 78.4; MS (EI) *m/z* (%) 322, 183, 166 (100), 152, 139, 77.

*Benzhydryl 2-bromobenzoate* (**3h**) [24]: white solid, 83% yield (61.0 mg), ^1^H NMR (400 MHz, CDCl_3_, ppm) *δ* 7.96 (t, *J* = 7.2 Hz, 1H), 7.10–7.48 (m, 13H), 7.07 (s, 1H); ^13^C NMR (100 MHz, CDCl_3_, ppm) *δ* 165.1, 139.9, 134.6, 132.8, 132.0, 131.6, 128.6, 128.1, 127.3, 127.2, 122.0, 78.5. MS (EI) *m/z* (%) 366, 183, 167 (100), 152, 77.

*Benzhydryl 2-methoxybenzoate* (**3i**) [24]: colorless liquid, 70% yield (44.6 mg), ^1^H NMR (400 MHz, CDCl_3_, ppm) *δ* 7.91 (d, *J* = 7.6 Hz, 1H), 7.28–7.50 (m, 11H), 7.10 (s, 1H), 6.97–7.00 (m, 2H); ^13^C NMR (100 MHz, CDCl_3_, ppm) *δ* 165.1, 159.6, 140.6, 133.8, 132.0, 128.5, 127.8, 127.3, 126.6, 120.2, 112.2, 77.3, 56.0. MS (EI) *m/z* (%) 318, 183, 167 (100), 152, 135, 77.

*Benzhydryl 2,4,6-trimethylbenzoate* (**3j**): colorless liquid, 85% yield (56.2 mg), ^1^H NMR (400 MHz, CDCl_3_, ppm) *δ* 7.30–7.43 (m, 10H), 7.17 (s, 1H), 6.84 (s, 2H), 2.29 (s, 3H), 2.17 (s, 6H); ^13^C NMR (100 MHz, CDCl_3_, ppm) *δ* 169.3, 140.1, 139.4, 135.2, 130.9, 128.5, 128.4, 128.0, 127.4, 77.5, 21.2, 19.7. MS (EI) *m/z* (%) 330, 167 (100), 155, 147, 77; HRMS calcd. for: C_23_H_22_NaO_2_ [M + Na]^+^: 353.1514, found *m/z* 353.1509.

*Benzhydryl 3-methylthiophene-2-carboxylate* (**3k**): colorless liquid, 88% yield (54.3 mg), ^1^H NMR (400 MHz, CDCl_3_, ppm) *δ* 7.30–7.46 (m, 11H), 7.07 (s, 1H), 6.94 (d, *J* = 4.8 Hz, 1H), 2.59 (s, 3H); ^13^C NMR (100 MHz, CDCl_3_, ppm) *δ* 161.7, 146.7, 140.4, 131.8, 130.3, 128.6, 127.9, 127.1, 126.7, 77.2, 16.0; MS (EI) *m/z* (%) 308, 167 (100), 152, 125, 77; HRMS calcd. for: C_19_H_16_NaO_2_S [M + Na]^+^: 331.0765, found *m/z* 331.0768.

*Benzhydryl cinnamate* (**3l**) [25]: white solid, 82% yield (51.6 mg), ^1^H NMR (400 MHz, CDCl_3_, ppm) *δ* 7.76 (d, *J* = 16.0 Hz, 1H), 7.54 (m, 2H), 7.30–7.39 (m, 13H), 7.02 (s, 1H), 6.57 (d, *J* = 16.4 Hz, 1H); ^13^C NMR (100 MHz, CDCl_3_, ppm) *δ* 166.0, 145.4, 140.3, 134.4, 130.4, 128.9, 128.6, 128.2, 127.9, 127.2, 118.1, 77.1; MS (EI) *m/z* (%) 314, 268, 236, 167 (100), 131, 77.

*Benzhydryl formate* (**3m**) [26]: colorless liquid, 80% yield (34.0 mg), ^1^H NMR (400 MHz, CDCl_3_, ppm) *δ* 8.24 (s, 1H), 7.26–7.36 (m, 10H), 7.01 (s, 1H); ^13^C NMR (100 MHz, CDCl_3_, ppm) *δ* 159.9, 139.6, 128.6, 128.1, 127.2, 76.6. MS (EI) *m/z* (%) 212, 184, 166 (100), 152, 77.

*Benzhydryl acetate* (**3n**) [17]: colorless liquid, 93% yield (42.1 mg), ^1^H NMR (400 MHz, CDCl_3_, ppm) *δ* 7.28–7.35 (m, 10H), 6.89 (s, 1H), 2.16 (s, 3H); ^13^C NMR (100 MHz, CDCl_3_, ppm) *δ* 170.0, 140.3, 128.5, 127.9, 127.2, 76.9, 21.3. MS (EI) *m/z* (%) 226, 184, 165 (100), 152, 105, 77.

*Benzhydryl pivalate* (**3o**) [22]: white solid, 83% yield (44.5 mg), ^1^H NMR (400 MHz, CDCl_3_, ppm) *δ* 7.30–7.36 (m, 10H), 6.84 (s, 1H), 1.27 (s, 9H); ^13^C NMR (100 MHz, CDCl_3_, ppm) *δ* 177.3, 140.7, 128.9, 127.8, 126.9, 76.6, 39.0, 27.2. MS (EI) *m/z* (%) 268, 211, 183, 167 (100), 152, 57.

*Benzhydryl hexanoate* (**3p**) [27]: colorless liquid, 82% yield (46.3 mg), ^1^H NMR (400 MHz, CDCl_3_, ppm) *δ* 7.30–7.36 (m, 10H), 6.90 (s, 1H), 2.43 (t, *J* = 7.4 Hz, 2H), 1.66–1.72 (m, 2H), 1.27–1.31 (m, 4H), 0.89 (t, *J* = 7.4 Hz, 3H); ^13^C NMR (100 MHz, CDCl_3_, ppm) *δ* 172.8, 140.5, 128.5, 127.9, 127.1, 76.7, 34.6, 31.3, 24.7, 22.3, 13.9. MS (EI) *m/z* (%) 282, 184, 166 (100), 152, 77.

*Phenyl(p-tolyl)methyl benzoate* (**3q**) [17]: white solid, 82% yield (49.6 mg), ^1^H NMR (400 MHz, CDCl_3_, ppm) *δ* 8.16 (d, *J* = 7.2 Hz, 2H), 7.60 (t, *J* = 7.2 Hz, 1H), 7.31–7.50 (m, 9H), 7.18 (d, *J* = 7.6 Hz, 2H), 7.11 (s, 1H), 2.35 (s, 3H); ^13^C NMR (100 MHz, CDCl_3_, ppm) *δ* 165.6, 140.5, 137.7, 137.5, 133.1, 130.5, 129.8, 129.3, 128.5, 128.4, 127.9, 127.2, 127.1, 77.4, 21.1. MS (EI) *m/z* (%) 302, 197, 180 (100), 165 (100), 105, 77.

*Phenyl(m-tolyl)methyl benzoate* (**3r**): colorless liquid, 80% yield (48.4 mg), ^1^H NMR (400 MHz, CDCl_3_, ppm) *δ* 8.17 (d, *J* = 7.6 Hz, 2H), 7.60 (t, *J* = 7.2 Hz, 1H), 7.27–7.50 (m, 10H), 7.11–7.13 (m, 2H), 2.36 (s, 3H); ^13^C NMR (100 MHz, CDCl_3_, ppm) *δ* 165.6, 140.5, 140.3, 138.2, 133.1, 130.5, 129.8, 128.8, 128.5, 128.5, 128.4, 127.9, 127.2, 124.2, 77.6, 21.5. MS (EI) *m/z* (%) 302, 197, 180, 165 (100), 105, 77. HRMS calcd. for: C_21_H_18_NaO_2_ [M]^+^: *m/z* 325.11990, found *m/z* 325.12047.

*Phenyl(o-tolyl)methyl benzoate* (**3s**): colorless liquid, 88% yield (53.2 mg), ^1^H NMR (400 MHz, CDCl_3_, ppm) *δ* 8.15 (d, *J* = 7.6 Hz, 2H), 7.22–7.61 (m, 13H), 2.39 (s, 3H); ^13^C NMR (100 MHz, CDCl_3_, ppm) *δ* 165.6, 139.6, 138.2, 136.0, 133.1, 130.7, 130.1, 129.8, 128.5, 128.4, 128.0, 127.9, 127.5, 127.2, 126.2, 75.0, 19.5; MS (EI) *m/z* (%) 302, 179 (100), 165, 105, 77; HRMS calcd. for: C_21_H_18_NaO_2_ [M + Na]^+^: 325.1199, found 325.1202.

*(4-Fluorophenyl)(phenyl)methyl benzoate* (**3t**): colorless liquid, 80% yield (49.0 mg), ^1^H NMR (400 MHz, CDCl_3_, ppm) *δ* 8.15 (d, *J* = 7.2 Hz, 2H), 7.61 (d, *J* = 7.0 Hz, 1H), 7.33–7.51 (m, 9H), 7.13 (s, 1H), 7.06 (t, *J* = 8.6 Hz, 2H); ^13^C NMR (100 MHz, CDCl_3_, ppm) *δ* 165.5, 162.5 (d, *J* = 245.4 Hz), 140.1, 136.2, 133.2, 132.7 (d, *J* = 9.0 Hz), 129.8, 129.1 (d, *J* = 8.0 Hz), 128.6, 128.5, 128.1, 127.0, 111.5 (d, *J* = 21.5 Hz), 76.8. MS (EI) *m/z* (%) 306, 201, 184 (100), 165, 105, 77. HRMS calcd. for: C_20_H_15_FNaO_2_ [M]^+^: *m/z* 329.09483, found *m/z* 329.09537.

*(2-Fluorophenyl)(phenyl)methyl benzoate* (**3u**): colorless liquid, 84% yield (51.5 mg), ^1^H NMR (400 MHz, CDCl_3_, ppm) *δ* 8.15 (d, *J* = 7.5 Hz, 2H), 7.47–7.60 (m, 6H), 7.28–7.37 (m, 5H), 7.15 (t, *J* = 7.4 Hz, 1H), 7.07 (t, *J* = 9.2 Hz, 1H); ^13^C NMR (100 MHz, CDCl_3_, ppm) *δ* 165.3, 160.1 (d, *J* = 246.8 Hz), 139.4, 133.2, 130.2, 129.8, 129.7 (d, *J* = 8.2 Hz), 128.6, 128.5, 128.1, 128.1, 127.9 (d, *J* = 13.2 Hz), 126.9, 124.3 (d, *J* = 3.6 Hz), 115.8 (d, *J* = 21.3 Hz), 71.8 (d, *J* = 2.9 Hz). MS (EI) *m/z* (%) 306, 184 (100), 165, 105, 77. HRMS calcd. for: C_20_H_15_FNaO_2_ [M]^+^: *m/z* 329.09483, found *m/z* 329.09526.

*(4-Chlorophenyl)(phenyl)methyl benzoate* (**3v**): colorless liquid, 78% yield (50.4 mg), ^1^H NMR (400 MHz, CDCl_3_, ppm) *δ* 8.13 (d, *J* = 7.6 Hz, 2H), 7.59 (t, *J* = 7.2 Hz, 1H), 7.31–7.49 (m, 11H), 7.08 (s, 1H); ^13^C NMR (100 MHz, CDCl_3_, ppm) *δ* 165.5, 139.9, 138.9, 133.9, 133.3, 131.5, 129.8, 128.8, 128.7, 128.6, 128.5, 128.2, 127.1, 76.8. MS (EI) *m/z* (%) 322, 200, 165 (100), 105, 77.

*(4-Tert-butylphenyl)(phenyl)methyl benzoate* (**3w**): colorless liquid, 76% yield (52.4 mg), ^1^H NMR (400 MHz, CDCl_3_, ppm) *δ* 8.15 (d, *J* = 7.5 Hz, 2H), 7.57 (t, *J* = 7.1 Hz, 1H), 7.44–7.47 (m, 4H), 7.29–7.37 (m, 7H), 7.11 (s, 1H), 1.30 (s, 9H); ^13^C NMR (100 MHz, CDCl_3_, ppm) *δ* 165.6, 150.9, 140.5, 137.3, 133.0, 130.5, 129.8, 128.5, 128.4, 127.8, 127.1, 127.0, 125.5, 77.4, 34.6, 31.3; MS (EI) *m/z* (%) 344, 239, 222 (100), 207, 105, 77; HRMS calcd. for: C_24_H_24_NaO_2_ [M + Na]^+^: 367.1669, found 367.1669.

*(4-Chlorophenyl)(p-tolyl)methyl benzoate* (**3x**): colorless liquid, 72% yield (48.5 mg), ^1^H NMR (400 MHz, CDCl_3_, ppm) *δ* 8.12 (d, *J* = 7.5 Hz, 2H), 7.58 (t, *J* = 7.2 Hz, 1H), 7.46 (t, *J* = 7.6 Hz, 2H), 7.31–7.37 (m, 6H), 7.17 (d, *J* = 7.7 Hz, 2H), 7.05 (s, 1H), 2.34 (s, 3H); ^13^C NMR (100 MHz, CDCl_3_, ppm) *δ* 165.5, 139.1, 138.0, 136.9, 133.8, 133.2, 130.2, 129.8, 129.4, 128.7, 128.5, 128.5, 127.1, 76.7, 21.1. MS (EI) *m/z* (%) 336, 214 (100), 179 (100), 165, 105, 77. HRMS calcd. for: C_21_H_17_ClNaO_2_ [M]^+^: *m/z* 359.08093, found *m/z* 359.08145.

*(4-Nitrophenyl)(phenyl)methyl benzoate* (**3y**): colorless liquid, 40% yield (26.7 mg), ^1^H NMR (400 MHz, CDCl_3_, ppm) *δ* 8.22 (d, *J* = 8.5 Hz, 2H), 8.14 (d, *J* = 7.5 Hz, 2H), 7.61 (d, *J* = 8.4 Hz, 3H), 7.49 (t, *J* = 7.6 Hz, 2H), 7.34–7.44 (m, 5H), 7.16 (s, 1H).^13^C NMR (100 MHz, CDCl_3_, ppm) *δ* 165.3, 147.4, 138.9, 133.5, 130.7, 130.1, 129.8, 128.9, 128. 7, 128.6, 127.8, 127.3, 123.9, 76.5. MS (EI) *m/z* (%) 333, 211, 165, 105 (100), 77.

*N-Benzhydrylbenzamide* (**4a**) [28]: white solid, 62% yield (35.6 mg), ^1^H NMR (400 MHz, CDCl_3_, ppm) *δ* 7.85 (d, *J* = 7.6 Hz, 2H), 7.54 (t, *J* = 7.2 Hz, 1H), 7.47 (t, *J* = 7.4 Hz, 2H), 7.32–7.40 (m, 10H), 6.68 (d, *J* = 6.8 Hz, 1H), 6.48 (d, *J* = 7.6 Hz, 1H); ^13^C NMR (100 MHz, CDCl_3_, ppm) *δ* 166.6, 141.5, 134.3, 131.7, 131.0, 128.8, 128.7, 127.6, 127.1, 57.5. MS (EI) *m/z* (%) 287, 182, 165, 105 (100), 77.

*N-Benzhydrylbenzenesulfonamide* (**4b**) [29]: white solid, 76% yield (49.2 mg), ^1^H NMR (400 MHz, CDCl_3_, ppm) *δ* 7.70 (d, *J* = 7.6 Hz, 2H), 7.49 (t, *J* = 7.4 Hz, 1H), 7.36 (t, *J* = 7.6 Hz, 2H), 7.11–7.23 (m, 10H), 5.63 (d, *J* = 7.2 Hz, 1H), 5.11 (d, *J* = 6.8 Hz, 1H); ^13^C NMR (100 MHz, CDCl_3_, ppm) *δ* 140.4, 132.4, 131.0, 128.8, 128.6, 127.7, 127.4, 127.2, 61.4. MS (EI) *m/z* (%) 322, 246, 182 (100), 167, 104, 77.

*N-Benzhydryl-4-methylbenzenesulfonamide* (**4c**) [29]: white solid, 70% yield (47.2 mg), ^1^H NMR (400 MHz, CDCl_3_, ppm) *δ* 7.50 (d, *J* = 8.0 Hz, 2H), 7.04–7.20 (m, 12H), 5.51 (d, *J* = 6.8 Hz, 1H), 4.94 (d, *J* = 6.4 Hz, 1H), 2.32 (s, 3H); ^13^C NMR (100 MHz, CDCl_3_, ppm) *δ* 143.1, 140.7, 137.6, 129.3, 128.5, 127.5, 127.4, 127.2, 61.4, 21.4. MS (EI) *m/z* (%) 336, 182 (100), 167, 91, 77.

*N-Benzhydrylmethanesulfonamide* (**4d**) [30]: white solid, 67% yield (35.0 mg), ^1^H NMR (400 MHz, CDCl_3_, ppm) *δ* 7.36–7.41 (m, 10H), 5.79 (d, *J* = 6.8 Hz, 1H), 4.99 (d, *J* = 6.4 Hz, 1H), 2.30 (s, 3H); ^13^C NMR (100 MHz, CDCl_3_, ppm) *δ* 140.7, 128.9, 128.0, 127.5, 61.3, 41.9. MS (EI) *m/z* (%) 259, 180 (100), 165, 104, 77.

## 4. Conclusions

In summary, we have developed a DDQ-promoted esterification and amination of benzylic C–H bonds under metal- and iodide-free conditions. A close to equal amount of coupling reagents is enough to afford the product in good to high yields. Functional groups, such as methyl, methoxy, fluoro, chloro, and bromo, were all well tolerated under the optimized reaction conditions. This method affords an efficient alternative route for the synthesis of esters and amides. The scope, mechanism, and synthetic applications of this reaction are under investigation.

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
