# Peer review of "An Efficient Metal-Free Oxidative Esterification and Amination of Benzyl C–H Bond"

_molecules, 2020, doi:10.3390/molecules25071527_

Round 1
Reviewer 1 Report
The authors have reported esterification and amination of benzylic C-H bonds by using DDQ under metal- and iodide-free conditions. The manuscript suffers from several weaknesses:
- Did any of the by-products generate from the reaction between the diphenylmethane and benzoic acid? i.e. diphenylmethanone.
- The authors are encouraged to show the conversion of diphenylmethane in table 1.
- In table 3, no product was obtained when 1-benzyl-4-nitrobenzene was used. The authors are encouraged to show more results from the reaction of diarylmethane with strong electron-withdrawing groups.
- Is reaction time required for excellent yields from these various diarylmethanes?
- The authors should make this mechanism clear, and provide some evidence.
- There are some typos from the organic chemical structures, i.e. diphenylmethane (trivalent carbon), etc.
Author Response
Response to Reviewer 1 Comments
Point 1: Did any of the by-products generate from the reaction between the diphenylmethane and benzoic acid? i.e. diphenylmethanone.
Response 1: I greatly appreciate your insightful comments and suggestions. No other by-products were observed except for diphenylmethanone. A large amount of diphenylmethanone generated from the reaction between the diphenylmethane and benzoic acid under oxygen or air. Other reactions were performed under argon and only a trace amount of diphenylmethanones were observed.
Point 2: The authors are encouraged to show the conversion of diphenylmethane in table 1.
Response 2: When the reaction of diphenylmethane and benzoic acid was carried out under O2, by-product diphenylmethanone was obtained in 45% yield as determined by GC (Table 1, entry 15). When the reaction was carried out under air, by-product diphenylmethanone was obtained in 28% yield as determined by GC (Table 1, entry 16). I have specified in Table 1.
Point 3: In table 3, no product was obtained when 1-benzyl-4-nitrobenzene was used. The authors are encouraged to show more results from the reaction of diarylmethane with strong electron-withdrawing groups.
Response 3: First of all, I feel very sorry for my negligence in presenting the wrong data in the table 3. The desired product 3y was obtained in 40% yield when 1-benzyl-4-nitrobenzene was used. The product 3y was characterized by 1H-NMR, 13C-NMR and MS. Due to the impact of the novel coronavirus epidemic, I can’t show more results from the reaction of diarylmethane with strong electron-withdrawing groups in a short time.
Point 4: Is reaction time required for excellent yields from these various diarylmethanes?
Response 4: The yield of 3a increased slightly when the reaction time was extended to 16 h or 24 h.
Point 5: The authors should make this mechanism clear, and provide some evidence.
Response 5: To gather more information about the reaction mechanism, the radical scavenger 2,2,6,6-tetramethyl-piperidine-1-oxyl (TEMPO) was added. The reaction was completely inhibited by 2 equivalents of TEMPO. This indicates that the present reaction process may involve radical intermediates.
Point 6: There are some typos from the organic chemical structures, i.e. diphenylmethane (trivalent carbon), etc.
Response 6: I have modified the relevant mistake and the proposed reaction mechanism based on relevant literatures (Eur. J. Org. Chem. 2019, 5650-5655; ChemSusChem 2012, 5, 2143-2146).

Reviewer 2 Report
“An Efficient Metal Free Oxidative Esterification and Amination of Benzyl C-H Bond”
The author described DDQ mediated acetoxylation and amination. Wide substrate generality was described. However, the proposed mechanism shown in Scheme 1 seems to be too speculative. The alternative mechanism would be benzylic cation was fist generated from 1a followed by nucleophilic attack of 2a. Did the author observed dimer derived from 1a? The intermediate B would undergo decarboxylation rapidly. The author should conduct further several control experiment to proof the radical-radical coupling. I think further investigation is required to clarify the reaction mechanism.
Author Response
Response to Reviewer 2 Comments
Point 1: The alternative mechanism would be benzylic cation was fist generated from 1a followed by nucleophilic attack of 2a. Did the author observed dimer derived from 1a? The intermediate B would undergo decarboxylation rapidly. The author should conduct further several control experiment to proof the radical-radical coupling. I think further investigation is required to clarify the reaction mechanism.
Response 1: I greatly appreciate your insightful comments and suggestions. No dimer derived from 1a was observed in this reaction. To gather more information about the reaction mechanism, the radical scavenger 2,2,6,6-tetramethyl-piperidine-1-oxyl (TEMPO) was added. The reaction was completely inhibited by 2 equivalents of TEMPO. This indicates that the present reaction process may involve radical intermediates. I have modified the proposed reaction mechanism based on relevant literatures (Eur. J. Org. Chem. 2019, 5650-5655; ChemSusChem 2012, 5, 2143-2146).

Reviewer 3 Report
The paper presented the advantages of applying DDQ for efficient esterification and amination of diarylmethanes under metal free conditions. I think it contributes towards further alternatives to easily obtain diarylmethane substituted esters and amines. The desired products were obtained in moderate to good yields.
Overall my impression of the work is excellent. The study includes precise planning of the experiments and adequate methods for analysis of the results. However, I advise the authors to find a native English speaker to proofread the manuscript because there are some mistakes.
I believe this work is publishable in Molecules after a minor revision of the English.
Author Response
Response to Reviewer 3 Comments
Point 1: I advise the authors to find a native English speaker to proofread the manuscript because there are some mistakes.
Response 1: I greatly appreciate your insightful comments and suggestions. As English is not my native language, I am sorry for some improper expressions in the article. I have asked someone who is good at English to check to avoid mistakes.
Round 2
Reviewer 1 Report
The revised manuscript has been improved from its original version, and the authors have addressed all my concerns.
Reviewer 2 Report
“An Efficient Metal Free Oxidative Esterification and Amination of Benzyl C-H Bond”
The revised manuscript was well improved based on the reviewer’s comments.
Therefore, I think the present revised manuscript would be acceptable for the publication in molecules.